# Hepatoprotective Effect of Silymarin Herb in Prevention of Liver Dysfunction Using Pig as Animal Model

**DOI:** 10.3390/nu17203278

**Published:** 2025-10-18

**Authors:** Prarthana Sharma, Varun Asediya, Garima Kalra, Sharmin Sultana, Nihal Purohit, Kamila Kibitlewska, Wojciech Kozera, Urszula Czarnik, Krzysztof Karpiesiuk, Marek Lecewicz, Paweł Wysocki, Adam Lepczyński, Małgorzata Ożgo, Marta Marynowska, Agnieszka Herosimczyk, Elżbieta Redlarska, Brygida Ślaska, Krzysztof Kowal, Angelika Tkaczyk-Wlizło, Paweł Grychnik, Athul P. Kurian, Kaja Ziółkowska-Twarowska, Katarzyna Chałaśkiewicz, Katarzyna Kępka-Borkowska, Ewa Poławska, Magdalena Ogłuszka, Rafał R. Starzyński, Hiroaki Taniguchi, Chandra Shekhar Pareek, Mariusz Pierzchała

**Affiliations:** 1Department of Pig Breeding, Faculty of Animal BioEngineering, University of Warmia and Mazury in Olsztyn, 10-719 Olsztyn, Poland; prarthana.sharma@student.uwm.edu.pl (P.S.); kamila.kibitlewska@uwm.edu.pl (K.K.); wojciech.kozera@uwm.edu.pl (W.K.); czar@uwm.edu.pl (U.C.); krzysztof.karpiesiuk@uwm.edu.pl (K.K.); mlecew@uwm.edu.pl (M.L.); pawel.wysocki@uwm.edu.pl (P.W.); 2Department of Animal Biochemistry and Biotechnology, Faculty of Animal BioEngineering, University of Warmia and Mazury in Olsztyn, 10-719 Olsztyn, Poland; 3Institute of Veterinary Medicine, Department of Infectious, Invasive Diseases and Veterinary Administration, Faculty of Biological and Veterinary Sciences, Nicolaus Copernicus University, 87-100 Toruń, Poland; varun@doktorant.umk.pl (V.A.); garimakalra@doktorant.umk.pl (G.K.); 503646@doktorant.umk.pl (S.S.); nihalpurohit13@doktorant.umk.pl (N.P.); 4Centre for Modern Interdisciplinary Technologies, Nicolaus Copernicus University, 87-100 Toruń, Poland; 5Department of Physiology, Cytobiology and Proteomics, Faculty of Biotechnology and Animal Husbandry, West Pomeranian University of Technology in Szczecin, 71-270 Szczecin, Poland; adam.lepczynski@zut.edu.pl (A.L.); malgorzata.ozgo@zut.edu.pl (M.O.); marta.marynowska@zut.edu.pl (M.M.); agnieszka.herosimczyk@zut.edu.pl (A.H.); elzbieta_lichwiarska@zut.edu.pl (E.R.); 6Institute of Biological Bases of Animal Production, Faculty of Animal Sciences and Bioeconomy, University of Life Sciences in Lublin, 20-950 Lublin, Poland; brygida.slaska@up.lublin.pl (B.Ś.); krzysztof.kowal@up.lublin.pl (K.K.); angelika.tkaczyk@up.lublin.pl (A.T.-W.); pawel.grychnik@up.lublin.pl (P.G.); athul.kurian@up.lublin.pl (A.P.K.); kaja.ziolkowska@up.lublin.pl (K.Z.-T.); 7Department of Genomics and Biodiversity, Institute of Genetics and Animal Biotechnology of the Polish Academy of Sciences, 05-552 Jastrzębiec, Poland; k.chalaskiewicz@igbzpan.pl (K.C.); k.kepka@igbzpan.pl (K.K.-B.); e.polawska@igbzpan.pl (E.P.); m.ogluszka@igbzpan.pl (M.O.); r.starzynski@igbzpan.pl (R.R.S.); h.taniguchi@igbzpan.pl (H.T.); 8Department of Experimental Embryology, Institute of Genetics and Animal Biotechnology of the Polish Academy of Sciences, 05-552 Jastrzębiec, Poland; 9Department of Molecular Biology, Institute of Genetics and Animal Biotechnology of the Polish Academy of Sciences, 05-552 Jastrzębiec, Poland; 10African Genome Center, Mohammed VI Polytechnic University, UM6P, Lot 660, Hay Moulay Rachid, Ben Guerir 43150, Morocco

**Keywords:** silymarin, silibinin, flavonolignans, pigs (*Sus scrofa*), pharmacokinetics, hepatoprotection, liver dysfunction, oxidative stress, inflammation, nutraceuticals

## Abstract

Silymarin, a flavonolignan-rich extract of *Silybum marianum*, is widely recognized for its hepatoprotective potential. While rodent studies predominate, pigs (*Sus scrofa*) offer a more translationally relevant model due to their hepatic architecture, bile acid composition, and transporter expression, which closely resemble those of humans. This narrative review synthesises current evidence on the chemistry, pharmacokinetics, biodistribution, and hepatoprotective activity of silymarin in porcine models. Available studies demonstrate that when adequate intrahepatic exposure is achieved, particularly through optimised formulations, silymarin can attenuate oxidative stress, suppress inflammatory signalling, stabilise mitochondria, and modulate fibrogenic pathways. Protective effects have been reported across diverse porcine injury paradigms, including toxin-induced necrosis, ethanol- and diet-associated steatosis, metabolic dysfunction, ischemia–reperfusion injury, and partial hepatectomy. However, the evidence base remains limited, with few long-term studies addressing fibrosis or regeneration, and methodological heterogeneity complicates the comparison of data across studies. Current knowledge gaps in silymarin research include inconsistent chemotype characterization among plant sources, limited reporting of unbound pharmacokinetic parameters, and variability in histological scoring criteria across studies, which collectively hinder cross-study comparability and mechanistic interpretation. Advances in analytical chemistry, transporter biology, and formulation design are beginning to refine the interpretation of exposure–response relationships. Advances in analytical chemistry, transporter biology, and formulation design are beginning to refine the interpretation of exposure–response relationships. In parallel, emerging computational approaches, including machine-learning-assisted chemotype fingerprinting, automated histology scoring, and Bayesian exposure modeling, are being explored as supportive tools to enhance reproducibility and translational relevance; however, these frameworks remain exploratory and require empirical validation, particularly in modeling enterohepatic recirculation. Collectively, current porcine evidence supports silymarin as a context-dependent yet credible hepatoprotective agent, highlighting priorities for future research to better define its therapeutic potential in clinical nutrition and veterinary practice.

## 1. Introduction



**Key Messages**



Pigs’ approximate human hepatic exposure and microanatomy are better than rodents.Chemotype and exposure drive efficacy more than dose labels.Advanced formulations change hepatic AUC, which changes outcomes.AI tools now allow objective histology, chemotype QC, and exposure-matched dose translation.Negative and neutral results are essential and must be of first-class data.

Liver dysfunction develops through the combined influence of several factors, including metabolic overload, activation of innate immune responses, exposure to xenobiotics, and drug-induced injury [1,2,3,4,5]. Since these processes occur simultaneously, focusing on a single pathway rarely provides lasting therapeutic benefit. Therefore, when evaluating any hepatoprotective agent, it is essential to relate its proposed mechanism of action to its absorption, distribution, metabolism, and excretion profile. Interpretation should also be based on unbound drug concentrations in plasma, bile, and liver rather than just the administered dose [6,7]. Silymarin, an extract derived from milk thistle (*Silybum marianum*), is widely used in nutritional supplementation and phytotherapy. It is not a single active molecule but a complex mixture of flavonolignans (Figure 1, which summarizes its main biological effects). The mixture is primarily composed of the silibinin diastereomers, together with isosilibinin, silychristin, silydianin, and structurally related compounds derived from a taxifolin backbone [8]. The relative proportions of these components vary depending on natural source material and processing methods, which in turn influence their biological behaviour and therapeutic profile.

Composition governs dissolution and membrane permeation, the extent of binding to plasma proteins, the balance of phase-II conjugation (glucuronidation and sulphation), and the degree of enterohepatic recirculation, together determining how much pharmacologically accessible material reaches hepatocytes (Figure 2) [9]. By the oral route, bioavailability is low without formulation support; most circulating species are highly albumin-bound, and a substantial fraction appears as conjugates formed by UDP-glucuronosyltransferases and sulphotransferases [10]. Conjugates are secreted into bile and may be hydrolysed in the gut, producing secondary plasma peaks and considerable between-animal variability. Hepatic transport proteins further shape exposure: uptake at the sinusoidal membrane involves OATP1B and NTCP, while canalicular efflux proceeds via BSEP, MRP2, and BCRP. These determinants need to be considered when interpreting efficacy signals [11,12].

Biological effects attributed to silymarin include activation of antioxidant defences through the Nrf2–ARE programme, for example via induction of haem oxygenase-1, NAD(P)H:quinone oxidoreductase-1, and glutamate–cysteine ligase. It also attenuates NF-κB-centred inflammatory pathways, with lower levels of tumour necrosis factor-α, interleukin-1β, and interleukin-6. In parallel, silymarin moderates TGF-β/SMAD signalling, causing shifts in the matrix metalloproteinase/tissue-inhibitor balance and reduced expression of α-smooth muscle actin and collagen. Improved lipid handling has also been reported through AMPK–PPAR circuits [13,14,15,16,17,18]. These effects are exposure-dependent and not equally evident; several derive from rodent studies conducted at supra-clinical exposures (Figure 3 and Appendix A). The key question for nutrition and therapeutic application is which of these effects persist in pigs at exposures attainable in humans [19].

This figure highlights the central molecular axis through which silymarin maintains hepatic homeostasis by balancing oxidative defense and inflammatory signalling. On the antioxidant side, silymarin activates the Nrf2/ARE pathway, which enhances transcription of phase II enzymes such as heme oxygenase-1, glutathione peroxidase, superoxide dismutase, and catalase, thereby supporting the glutathione pool and limiting reactive oxygen species [13,20,21]. In parallel, it suppresses the NF-κB pathway by preventing IκB degradation and nuclear translocation, leading to reduced transcription of pro-inflammatory mediators such as TNF-α, IL-1β, IL-6, COX-2, and iNOS [15,20]. The convergence of these pathways results in lower oxidative stress, decreased cytokine-driven injury, and protection against apoptosis and necrosis. This coordinated modulation of redox and inflammatory signalling is increasingly recognized as a defining mechanism underlying silymarin’s hepatoprotective action.

In vivo porcine and rodent studies have shown that silymarin treatment reduces hepatic lipid peroxidation, restores the glutathione redox balance (GSH/GSSG), and downregulates key pro-inflammatory mediators (TNF-α, IL-6, and NF-κB pathway components) [14,22,23,24,25]. This bidirectional regulation between oxidative stress and inflammation constitutes a system-level resilience mechanism that sets silymarin apart from other polyphenolic agents, which often demonstrate more limited, unidirectional antioxidant effects.

Porcine models are uniquely suited for investigating hepatoprotective mechanisms relevant to humans. Their lobular microarchitecture, sinusoidal hemodynamics, bile-acid profile, and the repertoire of hepatic uptake and efflux transporters collectively emulate the human liver more closely than standard rodent models (Table 1). These anatomical and biochemical congruencies underpin the translational fidelity of pigs in evaluating silymarin’s pharmacodynamics, bioavailability, and tissue-specific metabolic responses [10]. High-fat and high-cholesterol dietary regimens in pigs elicit metabolic derangements that reproduce the histopathological and molecular hallmarks of human non-alcoholic fatty liver disease (NAFLD), steatohepatitis (NASH), and metabolic syndrome. These include diffuse hepatic steatosis, mitochondrial oxidative stress, insulin resistance, and a pro-inflammatory cytokine milieu dominated by TNF-α and IL-6 signalling. Complementarily, surgical and ischemia–reperfusion paradigms, such as partial hepatectomy, provide controlled models for analysing biliary kinetics, microvascular congestion, and parenchymal hypoxia under conditions that parallel clinical hepatic injury. Collectively, these porcine frameworks capture the multifactorial pathophysiology of human liver dysfunction and offer a translationally relevant platform for evaluating hepatoprotective candidates such as silymarin, whose efficacy depends on coordinated modulation of redox balance, mitochondrial integrity, and inflammatory tone. Important differences persist in transport kinetics, and immune set-points can shift results; however, these require disciplined design, including matched controls, chemotype fingerprinting, and concurrent pharmacokinetics [20,26].

This narrative review synthesizes findings from porcine studies evaluating silymarin and its key constituents, such as silibinin, across toxic, dietary or ethanol-induced, ischemia–reperfusion, and surgical liver injury models. We also present a comparative overview of nutraceutical formulations currently in use, including native extracts, phospholipid complexes, self-emulsifying delivery systems, micelles, solid dispersions, cyclodextrin complexes, and nanosuspensions. Biochemical, histological, and molecular outcomes are interpreted alongside available pharmacokinetic data, with particular emphasis on unbound plasma concentrations where reported. Importantly, neutral or negative outcomes observed at exposure levels achievable in humans are considered as meaningful for decision-making. Human relevance is discussed only where exposure-matched translation is demonstrated.

## 2. Methods

This review was conducted as a structured narrative synthesis, focusing on porcine (*Sus scrofa*) studies that examined the pharmacokinetics and hepatoprotective effects of silymarin or its defined flavonolignan constituents, particularly silibinin. Relevant publications were identified through MEDLINE, Embase, and Web of Science up to September 2025 using combinations of terms for “silymarin,” “silibinin,” “milk thistle,” “pig,” “porcine,” “swine,” and “liver/hepatic,” with additional articles located by screening reference lists of retrieved papers. Controlled studies in pigs were the primary focus of inclusion. Each eligible report was required to specify the source or composition of silymarin—detailing the supplier, batch or lot number, or analytical fingerprint—and to include at least one biochemical indicator of hepatic function (e.g., ALT, AST, GLDH, ALP, GGT, bilirubin, or lipid parameters) together with histological or molecular evidence of hepatic injury such as steatosis, oxidative stress indices, cytokine expression, fibrosis, or stellate cell activation. While the synthesis emphasizes porcine data, mechanistic insights from complementary in vitro and rodent studies are referenced where necessary to interpret underlying pathways. Reports lacking essential methodological information were excluded.

From each eligible study, we extracted details of the experimental model (toxin challenge, diet or ethanol exposure, ischaemia–reperfusion, surgical or metabolic paradigm), the dosing strategy (prophylaxis or treatment), and the nature of the intervention, including compound identity, route of administration, formulation type where stated, and dose. Reported outcomes were collated across biochemical, histological, and molecular domains. Where pharmacokinetic data were available, we noted whether they referred to aglycones, conjugates, total concentrations, unbound plasma fractions, or hepatic tissue levels.

Given the heterogeneity of injury models, formulations, and outcome measures, a formal meta-analysis was not attempted. Instead, studies were compared qualitatively, with greater weight given to those that directly linked pharmacokinetic profiles with outcomes and clearly described extract composition or formulation. Where apparent benefits could plausibly be attributed to formulation artefacts or altered clearance rather than intrinsic hepatocellular protection, these were noted. Methodological limitations—such as incomplete reporting, absence of unbound pharmacokinetic data, or unstandardised histology—were explicitly considered when assessing strength of evidence. All in vivo studies included in this review were conducted under institutional or national ethical frameworks for animal research, as reported by the respective authors. Each study adhered to accepted principles of animal welfare, including the use of anaesthesia, humane endpoints, and approved ethical oversight to minimize pain or distress. Only studies explicitly citing compliance with recognized animal care guidelines were considered eligible for inclusion.

## 3. Silymarin Composition and Delivery

Silymarin is a complex mixture of flavonolignans, dominated by silibinin but also including Isosilybin, Silychristin, and Silydianin, each with distinct structural and pharmacological features (Table 2).

This table summarizes the major and minor flavonolignans present in silymarin extract, including their chemical identity, relative abundance, and pharmacological relevance. Silibinin (silybin) represents the dominant active component, contributing approximately 50–70% of total silymarin and serving as the principal hepatoprotective and antioxidant agent. The remaining components—Isosilybin, Silychristin, Isosilychristin, and Silandrin—occur in smaller proportions but contribute to the overall antioxidant, anti-inflammatory, and antifibrotic activity profile. Chemical formulas and molecular weights are provided for structural reference, and pharmacological functions are summarized based on in vitro and in vivo studies.

The major hepatoprotective and gastroprotective mechanisms of silymarin involve antioxidant, anti-inflammatory, antifibrotic, and cytoprotective pathways that converge to preserve hepatic and intestinal integrity (Figure 4). In pigs, silymarin displays three characteristic exposure regimes that explain much of the variation observed across studies. With native extracts, a solubility-limited regime predominates: dissolution is incomplete, precipitation is rapid, and plasma levels remain low, with aglycone fractions especially small [34,35,36].

This schematic outlines the major molecular pathways through which silymarin exerts protective effects on the liver and gastrointestinal tract. In hepatocytes, silymarin activates the Nrf2/ARE axis, enhances glutathione synthesis, and suppresses lipid peroxidation, thereby maintaining redox homeostasis. It attenuates pro-inflammatory signalling by inhibiting NF-κB and MAPK cascades, reduces TGF-β1–driven stellate cell activation, and limits fibrosis through downregulation of α-SMA and extracellular matrix deposition. At the mitochondrial level, silymarin stabilizes membrane potential, prevents cytochrome c release, and reduces caspase-mediated apoptosis. In the gastrointestinal tract, silymarin protects the gastric mucosa by enhancing prostaglandin synthesis, nitric oxide release, and mucin secretion, while counteracting oxidative stress and inflammatory cytokine production. Together, these actions contribute to preserved hepatocyte viability, reduced fibrosis and necrosis, and strengthened mucosal defence, highlighting silymarin’s dual protective role in hepatic and gastrointestinal tissues.

Any pharmacological signal under these conditions often depends on enterohepatic recycling to prolong contact with the liver. A second regime is observed when bile flow or sinusoidal transport is perturbed, as occurs in cholestasis, ischemia–reperfusion, or after the administration of surfactant-rich formulations [37]. Here, total plasma exposure may rise because canalicular export is restricted, but hepatocellular delivery falls. This produces apparently improved enzyme profiles without genuine hepatoprotection [10,36]. The third regime emerges with well-designed formulations that succeed in raising unbound aglycone at the portal inlet [38,39]. Under these conditions, peak concentrations are higher, hepatic enrichment improves, and downstream redox and inflammatory responses become detectable, provided that dosing is timed appropriately with the onset of injury. Identifying which of these regimes a study represents is more meaningful than simply comparing nominal milligram doses [40,41].

Formulation design remains the most experimentally manageable variable governing silymarin bioavailability, yet its pharmacological interpretation must rely on unbound rather than total systemic exposure. Among advanced delivery systems, phospholipid complexes and self-emulsifying formulations consistently enhance intestinal absorption and hepatic enrichment by promoting membrane partitioning and partial lymphatic routing. In porcine models, these systems yield markedly higher unbound plasma and hepatic concentrations than conventional powdered extracts, aligning with the species’ robust bile-acid turnover and mixed conjugation profile. In contrast, formulations with excessive surfactant or bile-salt mimetic content can transiently inhibit key hepatic transporters such as OATP1B1/1B3 and BSEP, leading to apparent biochemical “improvements” that may instead reflect reduced hepatocellular uptake. Solid dispersions, nanosuspensions, and cyclodextrin complexes can also enhance apparent absorption when physicochemical stability is maintained, although their benefit diminishes if unbound exposure does not increase proportionally. Collectively, these findings emphasise that formulation success should be assessed not by total plasma levels but by free drug delivery to hepatocytes—the true pharmacodynamic currency underpinning hepatoprotection [42].

## 4. Porcine Liver Injury and Surgical Models

### 4.1. Carbon Tetrachloride and Related Toxins

Carbon tetrachloride (CCl_4_) remains a classical hepatotoxicant, metabolised by CYP2E1 to trichloromethyl radicals that initiate centrilobular lipid peroxidation and necrosis [42]. Thioacetamide evokes a similar oxidative injury pattern, progressing to fibrosis with chronic exposure, whereas acetaminophen (paracetamol) and valproate induce glutathione depletion and mitochondrial dysfunction, respectively. Within these intrinsic DILI paradigms, silymarin’s antioxidant and membrane-stabilising properties—mediated via Nrf2 activation and NF-κB suppression—are mechanistically plausible, provided sufficient aglycone exposure reaches hepatocytes [43]. Informative endpoints include serum aminotransferases, glutamate dehydrogenase (GLDH) for mitochondrial specificity, oxidative stress indices (MDA, GSH/GSSG), centrilobular histopathology, and cytokine profiling, ideally complemented by intrahepatic concentration measurements of both aglycones and conjugates [44].

### 4.2. Ethanol- and Diet-Associated Injuries

Ethanol intoxication enhances CYP2E1 activity, increases mitochondrial ROS production, and sensitises the liver to gut-derived endotoxins, thereby elevating TNF-α and IL-6. High-fat, cholesterol-rich or fructose-supplemented diets exacerbate steatosis, remodel bile-acid pools, and alter transporter expression, modulating the disposition of silymarin [45]. In these contexts, anti-inflammatory effects are dependent on the alignment of peak unbound exposure with binge or acute injury events, whereas redox stabilisation is driven by cumulative exposure. Key endpoints include aminotransferases, GLDH, bile flow, hepatic triglyceride content, cytokine panels, and histological evaluation of steatosis, ballooning, inflammation, and fibrosis [46].

### 4.3. Metabolic Dysfunction Models

High-fat or cholesterol diets in pigs produce macro vesicular steatosis, insulin resistance, and variable fibrosis, closely resembling human metabolic dysfunction–associated steatotic liver disease (MASLD). These models also reveal the diet-sensitivity of conjugation and transporter pathways, factors that critically determine silymarin exposure [46]. Mechanistically, sustained intrahepatic levels are expected to attenuate lipid peroxidation, stabilise glutathione indices, and reduce profibrotic signalling through TGF-β and α-smooth muscle actin. Informative markers include quantitative histology with collagen proportionate area or hydroxyproline content, oxidative stress indices, and transporter proteomics. Longitudinal studies of sufficient duration are necessary to capture fibrotic progression and therapeutic impact.

### 4.4. Ischaemia–Reperfusion Injury

Warm ischaemia during inflow clamping or cold preservation in transplant models induces microvascular collapse, sinusoidal congestion, and endothelial dysfunction, followed by a burst of oxidative and inflammatory activity upon reperfusion [13,47]. These insults are acutely peak-sensitive: protective effects of silymarin depend on the presence of C_max, free at or before reperfusion. Relevant readouts include early enzyme release (ALT, AST, LDH), bile-flow parameters, microcirculatory indices, and short-term histology (24–48 h). Rigor in controlling clamp time, temperature, and portal pressure is essential for comparability [48,49].

### 4.5. Partial Hepatectomy and Regeneration

Partial hepatectomy (60–70% resection) triggers portal hyper perfusion, cytokine priming via IL-6/STAT3, and a wave of hepatocyte proliferation [50]. Silymarin may stabilise mitochondria, mitigate oxidative and ER stress, and support regenerative signalling. Because regeneration unfolds over several days, cumulative intrahepatic exposure is more relevant than single-peak concentrations [51]. Informative endpoints include volumetric recovery by imaging, proliferation indices (Ki-67, PCNA), bile flow, and functional clearance tests such as indocyanine green.

### 4.6. Drug-Induced Liver Injury and Specialised Models

Intrinsic drug-induced liver injury (DILI) paradigms, typified by acetaminophen and valproate models, reproduce a reproducible pathogenic sequence: rapid depletion of reduced glutathione, collapse of mitochondrial membrane potential with impaired ATP synthesis, a surge in reactive oxygen species, and consequent hepatocellular necrosis or apoptosis. For mechanistic and translational clarity, experimental designs must explicitly define the therapeutic window for silymarin (pre-treatment/prophylaxis, co-administration/protection, or post-injury/therapeutic dosing) and align dosing with measured unbound exposure and injury kinetics. Reporting of time-resolved biomarkers (e.g., GSH/GSSG, mitochondrial integrity metrics, ATP levels, and peak ALT/AST) substantially improves interpretation by distinguishing antioxidant prevention from later restorative effects on cell survival and regeneration [52]. Transporter-stress or cholestatic models, induced by oestrogens, cyclosporine, or bosentan, interrogate BSEP-limited export, while amiodarone or tamoxifen introduces phospholipidosis [53]. Here, apparent benefits may arise from excipient-mediated transporter inhibition rather than direct hepatocellular protection, underscoring the need to quantify portal C_max, free against known transporter-inhibition thresholds. Bile-duct ligation represents a stringent model of cholestasis, better suited to testing the limits of hepatoprotective claims than establishing efficacy [54,55].

### 4.7. Methodological Considerations

Interpretation of porcine models is frequently undermined by recurrent methodological issues. Chemotype ambiguity highlights the need for LC–MS or quantitative NMR fingerprinting of each extract. Reliance on total plasma AUC, dominated by conjugates, overstates pharmacologically relevant exposure; unbound metrics provide a more accurate measure. Surfactant-based formulations risk confounding through transporter inhibition. Peak-driven insults, such as ischaemia, reperfusion demand precise dosing schedules to capture causality. Without bile-flow or clearance data, cholestasis-induced artefacts may masquerade as efficacy. Histological variability remains a concern, necessitating calibrated scoring or validated digital pipelines with inter-observer agreement reported. Finally, deficiencies in randomisation, blinding, pre-specified endpoints, and statistical power weaken claims and must be addressed in future studies.

### 4.8. Comparative Pharmacology of Silymarin Across Species

The principal mechanistic pathways through which silymarin and its flavonolignans exert hepatoprotective effects—including antioxidant buffering, anti-inflammatory signalling, and antifibrotic modulation—are summarised in Table 3.

Any serious appraisal of silymarin as a hepatoprotective agent must grapple with the fact that its pharmacological behaviour is profoundly species-dependent [52,80]. Rodent models, though invaluable for mechanistic exploration, often generate signals that are exaggerated in magnitude and distorted in character when compared to human outcomes.

### 4.9. Oral Absorption and First-Pass Handling

Across experimental species—rodents, pigs, and humans—silymarin demonstrates a shared pharmacokinetic constraint: inherently low oral absorption driven by poor aqueous solubility, rapid phase II conjugation, and extensive first-pass metabolism [13]. In rodents, oral bioavailability rarely exceeds 1%, constrained by rapid glucuronidation and sulphation, predominantly via Ugt1a and Sult1a isoforms. Achieving pharmacodynamic readouts in these settings typically requires doses far above anything used clinically (>200 mg/kg), meaning mechanistic clarity is often achieved at the cost of translational plausibility [81].

By contrast, pigs reproduce human oral pharmacokinetics more closely. Following conventional extract administration, peak plasma concentrations typically fall in the low nanomolar to sub-micromolar range, with 70–85% of circulating species existing as conjugates [82].

### 4.10. Plasma Binding and Circulating Fractions

All flavonolignans bind plasma proteins extensively (>95%, chiefly to albumin), but species differences in binding affinity alter the pharmacologically relevant free fraction [82]. Rodents exhibit marginally weaker binding, inflating the available unbound pool and artificially amplifying effect sizes in vivo [6]. Pigs and humans, however, display near-identical binding behaviour, such that free concentrations are both lower and more realistic. Since hepatoprotective activity depends on transient bursts of unbound aglycone reaching hepatocytes rather than the bulk of circulating conjugates, this alignment between pigs and humans is pivotal for credible dose–response translation [83].

### 4.11. Hepatic Transport and Cellular Entry

Transporter biology is arguably the most decisive factor in determining cross-species pharmacological relevance. In rodents, sinusoidal uptake is mediated mainly by Oatp1a1/1b2 and Ntcp, transporters whose substrate preferences differ significantly from the human orthologues OATP1B1/1B3 and NTCP. Efflux in rodents is dominated by Mrp2, with only limited contribution from Bcrp. In humans, however, uptake occurs through coordinated activity of OATP1B1/1B3 and NTCP, counterbalanced by efflux via BSEP, MRP2, and BCRP. Notably, pigs closely replicate this human transporter pattern, both in repertoire and kinetic behaviour [84]. As a result, pigs—unlike rodents—can exhibit clinically relevant pharmacological liabilities such as BSEP inhibition–associated cholestasis or OATP-mediated drug–drug interactions, both of which are critical when assessing translational potential.

### 4.12. Enterohepatic Cycling and Bile Flow

Silymarin’s multiphasic plasma profiles in pigs and humans reflect extensive enterohepatic recycling. Conjugated flavonolignans secreted in bile are hydrolysed in the gut and reabsorbed, producing secondary plasma peaks that sometimes surpass the primary [85,86]. This recycling increases total AUC, even though the free hepatocellular exposure remains relatively stable. In rodents, by contrast, the bile pool is dominated by muricholic acids and microbial hydrolysis is less efficient, which dampens recycling and produces a more monotonic decline in plasma concentration curves [41]. As a result, pigs represent the only large-animal model that accurately reproduces the oscillatory kinetic profile seen in humans. This distinction is clinically relevant: in peak-sensitive injury models, such as ischaemia–reperfusion, therapeutic benefit depends on whether drug exposure aligns with the timing of the oxidative burst.

### 4.13. Metabolism and Enzymatic Pathways

Rodents metabolise silymarin predominantly through phase II conjugation, with negligible cytochrome P450 involvement [87]. Humans, by contrast, show a more diversified enzymatic portfolio: UGT1A1 and UGT1A9, SULT1A1, and CYP3A4 all contribute. Pigs occupy an intermediate but human-like position [88]. They exhibit strong UGT activity alongside CYP3A-driven oxidation, yielding a metabolite spectrum far closer to that of humans than rodents. Moreover, both pigs and humans share inducibility of CYP2E1, the key driver of carbon tetrachloride- and ethanol-related hepatotoxicity, making them uniquely suitable for toxin-sensitive pharmacology that rodents often misrepresent [87,89].

### 4.14. Pharmacodynamics and Dose–Response Alignment

Rodent studies consistently report dramatic antioxidant and anti-inflammatory responses: strong Nrf2 activation, marked suppression of NF-κB, and clear downregulation of TGF-β/SMAD signalling. Yet these effects generally appear at exposures far above anything achievable in humans [42]. Pigs, in contrast, display more modest but mechanistically credible effects: reductions in aminotransferases, improved glutathione indices, and partial histological protection, all contingent on formulations that raise intrahepatic unbound concentrations [13,20]. Human clinical data align with this pattern: reductions in transaminases and oxidative stress markers are consistent, but never spectacular. Thus, porcine outcomes not only resemble human effect sizes but also capture the variability intrinsic to nutraceutical responses [90].

### 4.15. Safety, Tolerability, and Transporter Liabilities

Although silymarin has a wide safety margin across all models, species diverge sharply in their ability to manifest transporter-dependent toxicities [91]. Rodents rarely display BSEP or OATP inhibition, even at high doses, which risks underestimating formulation liabilities [92]. In pigs and humans, however, excipients or nanoparticle carriers can inhibit these transporters and precipitate cholestatic signatures, underlining why pigs are essential for preclinical safety assessment. This becomes especially important with next-generation delivery systems—self-emulsifying drug carriers, nanosuspensions, or solid dispersions—where excipient–transporter interactions may confound interpretations of efficacy and safety [93,94].

### 4.16. Translational Perspective

Viewed in aggregate, the comparative pharmacology of silymarin underscores a critical point: rodents are indispensable for dissecting molecular mechanisms, but they consistently overstate efficacy and underplay liability. Pigs, by contrast, replicate the human pharmacological landscape: low baseline bioavailability, extensive conjugation, enterohepatic recycling, and a transporter environment that both enables hepatocellular delivery and exposes safety margins. Their role, therefore, is not simply confirmatory but adjudicative—identifying which formulations deliver pharmacologically relevant exposure, distinguishing true hepatoprotection from clearance artefacts, and defining exposure–response windows that can be credibly extrapolated to humans [45,95].

By embedding porcine studies within a framework of chemotype fingerprinting, unbound pharmacokinetics, and transporter-aware endpoints, the field can move beyond anecdotal claims and establish a rigorous translational bridge. Without such discipline, silymarin research risks remaining trapped between rodent optimism and human ambiguity. With it, pigs can become the decisive model that links mechanistic plausibility to clinical credibility.

## 5. Evidence from Porcine Studies

Representative experimental and clinical studies illustrating the pharmacological properties and mechanisms of silymarin are summarized in Appendix A [15,21,25,96,97,98,99,100,101,102,103,104,105,106,107,108,109,110,111,112,113,114,115,116,117,118,119,120,121]. These data highlight antioxidant, anti-inflammatory, and immunomodulatory actions observed across species, but also reveal the need for translation into porcine systems. The existing porcine evidence base for silymarin remains limited but mechanistically revealing. Collectively, these studies confirm the feasibility of conducting pharmacologically controlled interventions in pigs, delineate the influence of formulation on systemic and hepatic exposure profiles, and expose a persistent translational discontinuity between elevated systemic concentrations and reproducible hepatocellular protection under defined injury paradigms. Bridging this disconnect will require integrated designs that link unbound pharmacokinetics, intracellular target engagement, and histological endpoints within harmonized experimental frameworks. The most rigorous pharmacokinetic data come from Xu et al. [10], who compared a solid-dispersion formulation of silybin with a conventional premix in sixteen healthy pigs. The solid dispersion increased peak plasma concentrations nearly threefold (≈1190 ng/mL vs. 411 ng/mL) and more than doubled systemic exposure (AUC ≈ 1299 vs. 587 ng·h/mL). This study convincingly demonstrated that formulation strategy dictates systemic bioavailability. However, because it was conducted in healthy animals without a liver-injury model, it did not establish whether enhanced exposure translates into measurable hepatoprotection.

Peripheral investigations in agricultural and cellular settings provide contextual but indirect evidence relevant to porcine hepatoprotection [122]. In production pigs, dietary inclusion of milk-thistle preparations has been associated with favourable growth performance and physiological tolerance, yet the absence of aminotransferase, bile-flow, or histological assessments limits any inference regarding hepatic protection. Complementary phytochemical and cell-culture studies reaffirm the antioxidant, antimicrobial, and metabolic-modulatory potential of *Silybum marianum* extracts and reveal tissue-specific regulation of xenobiotic pathways, particularly CYP1A1 and phase-II conjugation systems. While these data enrich the mechanistic backdrop, they remain several inferential steps removed from controlled in vivo liver-injury paradigms in pigs. Their chief value lies in defining plausible molecular entry points that future porcine studies can interrogate under standardized hepatotoxic challenges. For translational orientation, human pharmacokinetic and clinical data offer important benchmarks. Standardized extracts (70–80% silymarin) are typically administered at 100–300 mg three times daily [123]. Oral silibinin reaches peak plasma concentrations of ~200–1400 µg/L within two hours, with approximately 75% present as conjugates [124]. The elimination half-life is ~6 h, with 5–8% excreted renally, 20–40% via bile, and the remainder in faeces [90,125]. Biliary concentrations can exceed plasma by up to 100-fold, reflecting extensive enterohepatic recycling. Clinical trials, though heterogeneous, consistently demonstrate biological signals: reductions in transaminases in chronic active hepatitis [126], improved antioxidant enzyme activity in alcoholic liver disease [127], and modest reductions in lipoperoxidative injury in patients on long-term psychotropic therapy [128]. A recent meta-analysis of 55 randomized trials (*n* = 3545) concluded that silymarin significantly lowers AST and ALT, particularly in younger, non-obese patients with NAFLD or viral hepatitis, though benefits were less clear in alcohol- or drug-induced injury [129].

## 6. Microbiome and the Gut–Liver Axis

Silymarin’s trajectory does not end at absorption; the gut microbiome continues the story. In pigs and humans, flavonolignan conjugates excreted in bile are cleaved by bacterial β-glucuronidases and sulfatases, regenerating aglycones that re-enter the portal circulation and create the multiphasic plasma profiles absent in rodents [130]. These reactions also produce smaller phenolic metabolites with independent antioxidant and anti-inflammatory activity, widening the pharmacological footprint beyond the parent compounds. Evidence from porcine studies shows that these processes matter in vivo: sows supplemented with silymarin during late gestation and lactation displayed reproducible shifts in fecal community structure—including increases in *Fibrobacteres* and *Actinobacteria*—that correlated with lower circulating IL-1β and reduced systemic inflammation [10]. Feeding trials in growing and finishing pigs reinforce this picture, reporting increased *Lactobacillus* counts, reduced *Escherichia coli*, improved nutrient digestibility, and enhanced antioxidant enzyme activity, with micellar and solid-dispersion formulations achieving superior bioavailability compared with conventional premixes [10].

Together, these findings point to three interacting mechanisms: microbial deconjugation that elevates portal delivery of active aglycones; microbial metabolites that extend systemic antioxidant and immunoregulatory actions; and community shifts that stabilize barrier function and bile acid homeostasis, thereby reducing hepatic exposure to inflammatory insults. Because formulation can double Cmax and AUC, the pharmacological impact of microbiome recycling depends strongly on how silymarin is delivered. Addressing this interaction will require integrated porcine studies that combine unbound pharmacokinetics with longitudinal metagenomics, bile acid and metabolomic profiling, and causal microbiome perturbations such as fecal microbiota transplantation or targeted depletion. Only with this evidence can we resolve whether silymarin’s hepatoprotective effects are primarily driven by hepatocellular exposure, microbiome-mediated pathways, or their synergy [21].

## 7. Computational Adjuncts for Translational Rigor

Emerging computational tools may offer supportive value in strengthening experimental design and data interpretation within silymarin research. When applied as rigorously validated adjuncts to empirical investigation, machine-learning and image-analysis approaches can assist in chemotype standardization, objective histological scoring, and systematic synthesis of scattered datasets. In the context of porcine hepatoprotective models—where sample sizes are small and experimental heterogeneity is high—such tools may reduce analytical bias and enhance reproducibility. Their promise, however, remains conditional on transparent benchmarking against experimental results, reinforcing that computational analyses complement rather than replace the discipline of biological validation.

## 8. Formulation Science and Translational Pharmacology

The translational trajectory of silymarin remains defined by the challenge of overcoming its intrinsic pharmacokinetic limitations—poor aqueous solubility, extensive conjugation, and strong plasma-protein binding—which restrict the delivery of pharmacologically active, unbound molecules to hepatocytes (Figure 5).

Accordingly, recent advances have shifted focus from dose escalation to formulation design, seeking to enhance unbound hepatic exposure while preserving biochemical safety [29]. In porcine studies, phospholipid complexes, self-emulsifying systems, solid dispersions, and nanosuspensions have each achieved measurable gains in apparent bioavailability; yet formulation-dependent variability and the absence of consistent histological corroboration highlight that elevated systemic concentrations alone cannot confirm hepatocellular protection. Emerging nanotechnology-based carriers offer a conceptual path toward targeted hepatic delivery but remain largely invalidated in porcine hepatology. Collectively, these developments underscore a simple translational truth: formulation innovation can optimize exposure, but meaningful therapeutic progress depends on transporter-aware, histologically anchored validation that directly links pharmacokinetics to functional hepatic recovery.

Formulation choice must also be matched to disease dynamics. Peak-sensitive injuries such as ischaemia–reperfusion require carriers that deliver rapid unbound spikes now of reperfusion, while fibrogenic and metabolic models demand systems sustaining exposure over weeks to modulate stellate cell activation and matrix remodelling. Few porcine studies have aligned dosing schedules with these biological windows, yet such coupling is essential for credible translational claims [36].

From a regulatory standpoint, unbound area under the curve (AUC), C_max, free, and liver-to-plasma unbounded partitioning coefficient (Kpuu) are now considered more informative than nominal dose. Porcine models can uniquely define these parameters because their hepatobiliary transporters and bile acid pools approximate human conditions. Future progress depends on integrating chemotype fingerprinting, advanced formulations, and transporter-aware pharmacokinetics into well-controlled porcine injury paradigms.

## 9. Conclusions

Porcine studies demonstrate that silymarin can reduce hepatocellular injury, modulate redox balance, and attenuate inflammatory responses when exposure is optimised through well-designed formulations or consistent supplementation. In toxin-challenged pig models, improvements in aminotransferase profiles, histological integrity, and mitochondrial function provide supportive large-animal evidence for a measurable protective effect. Comparable benefits observed in productive sows under physiological stress suggest practical nutritional relevance.

However, current evidence remains limited in scope. Most studies focus on acute injury, with little evaluation of fibrosis, regeneration, or long-term metabolic outcomes, and only a minority report unbound pharmacokinetics alongside mechanistic endpoints. Thus, while consistent redox and inflammatory improvements are evident, sustained structural protection has not yet been demonstrated.

Within these boundaries, the porcine model offers a translationally relevant framework for evaluating silymarin under exposure conditions comparable to human use. When controlled for formulation, dose, and timing, it provides a realistic platform for assessing hepatoprotective potential beyond rodent proof-of-concept studies.

## Figures and Tables

**Figure 1 nutrients-17-03278-f001:**
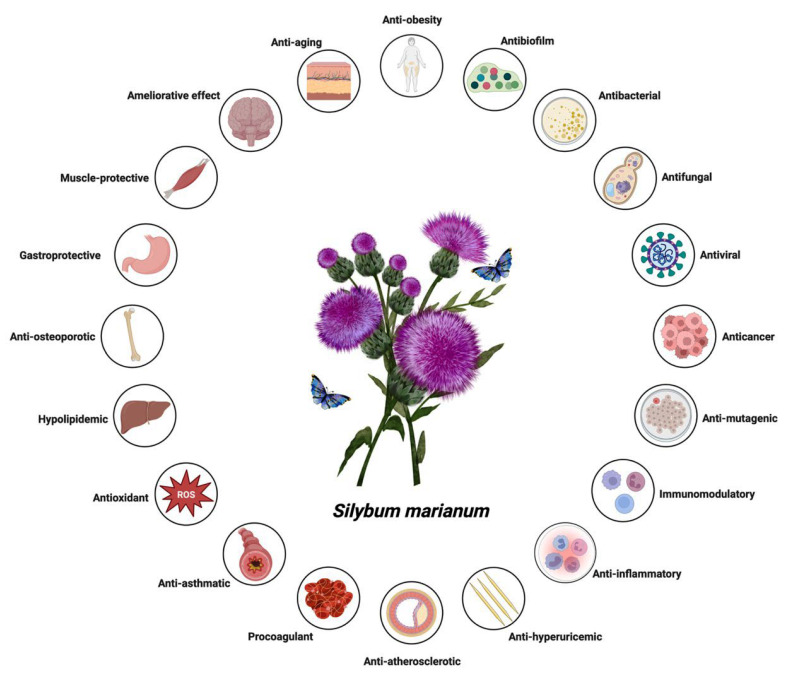
System-wide health benefits of *Silybum marianum* (silymarin). This schematic summarises the extensive spectrum of therapeutic and biological effects ascribed to *Silybum marianum*, a medicinal botanical abundant in flavonolignans (predominantly silymarin). The plant manifests antioxidant properties via the scavenging of reactive oxygen species (ROS), in addition to exhibiting anti-inflammatory, immunomodulatory, anti-mutagenic, anticancer, antiviral, antifungal, antibacterial, and antibiofilm activities. Furthermore, *S. marianum* reveals anti-obesity, anti-aging, ameliorative, muscle-protective, gastroprotective, anti-osteoporotic, hypolipidemic, anti-asthmatic, procoagulant, anti-atherosclerotic, and anti-hyperuricemic characteristics. Collectively, these pharmacological actions emphasize the plant’s multi-target therapeutic potential across metabolic, hepatic, cardiovascular, musculoskeletal, and immune systems.

**Figure 2 nutrients-17-03278-f002:**
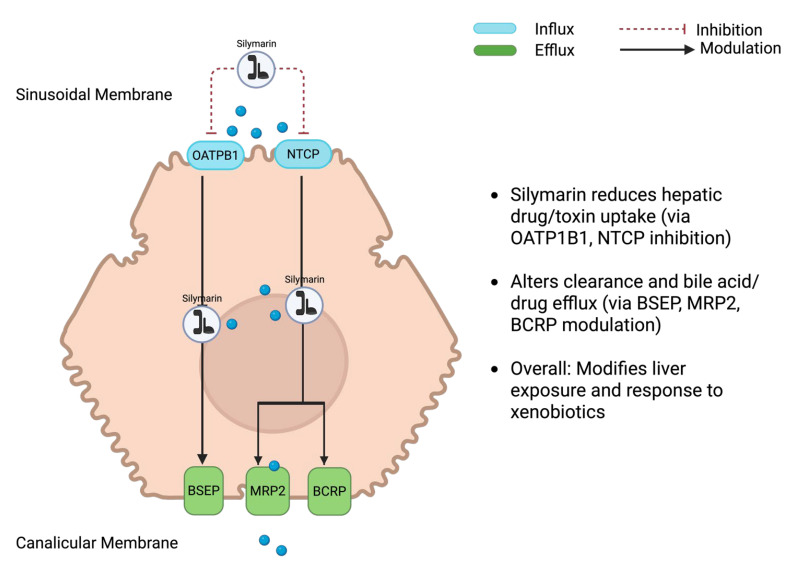
Silymarin and the regulation of hepatic transporter proteins. Silymarin is predominantly internalized by hepatocytes through the organic anion transporting polypeptide 1B1 (OATP1B1) and the sodium taurocholate co-transporting polypeptide (NTCP) situated on the basolateral membrane. Subsequent to its uptake, silymarin, along with its conjugated metabolites (blue balls) are expelled into the bile via canalicular efflux transporters, which encompass the bile salt export pump (BSEP), multidrug resistance-associated protein 2 (MRP2), and breast cancer resistance protein (BCRP). The dashed red lines denote the inhibitory modulation of transporter functionality, whereas solid arrows illustrate the trajectory of compound movement. Collectively, these transport mechanisms govern the hepatic disposition, bioavailability, and pharmacokinetic characteristics of silymarin.

**Figure 3 nutrients-17-03278-f003:**
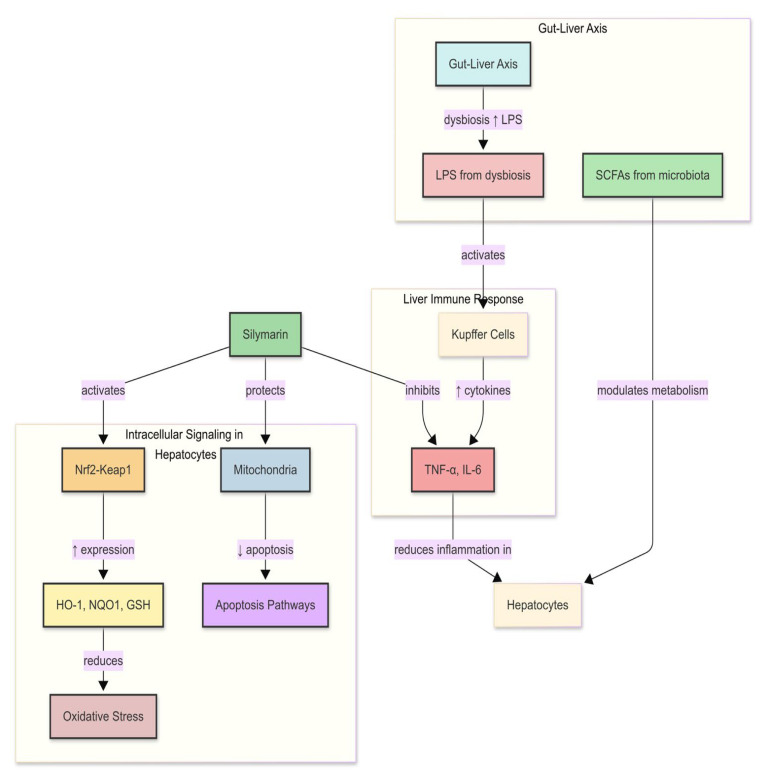
Crosstalk between antioxidant and inflammatory pathways modulated by silymarin in hepatocytes. This schematic delineates the multifaceted protective mechanisms of silymarin in preserving hepatic homeostasis. Within hepatocytes, silymarin initiates the activation of the Nrf2–Keap1 signaling pathway, resulting in the upregulation (↑) of genes associated with antioxidant defense, including HO-1, NQO1, and GSH, thus alleviating oxidative stress. Simultaneously, it safeguards mitochondrial integrity, thereby diminishing apoptosis. In the context of the hepatic immune microenvironment, silymarin hinders the activation of Kupffer cells and downregulates (↓) proinflammatory cytokines (TNF-α, IL-6), which collectively serve to mitigate liver inflammation. At the systemic level, via the gut–liver axis, dysbiosis of the gut microbiota elevates levels of lipopolysaccharides (LPSs), which incite hepatic immune responses, whereas short-chain fatty acids (SCFAs) produced by a healthy microbiota contribute to metabolic equilibrium. Silymarin indirectly antagonizes LPS-induced inflammation and modulates metabolic homeostasis through gut–liver interactions. Collectively, these actions exemplify the dual regulatory capacity of silymarin in cellular redox defense and immunometabolic equilibrium, thereby underscoring its therapeutic promise in the management of chronic liver disease.

**Figure 4 nutrients-17-03278-f004:**
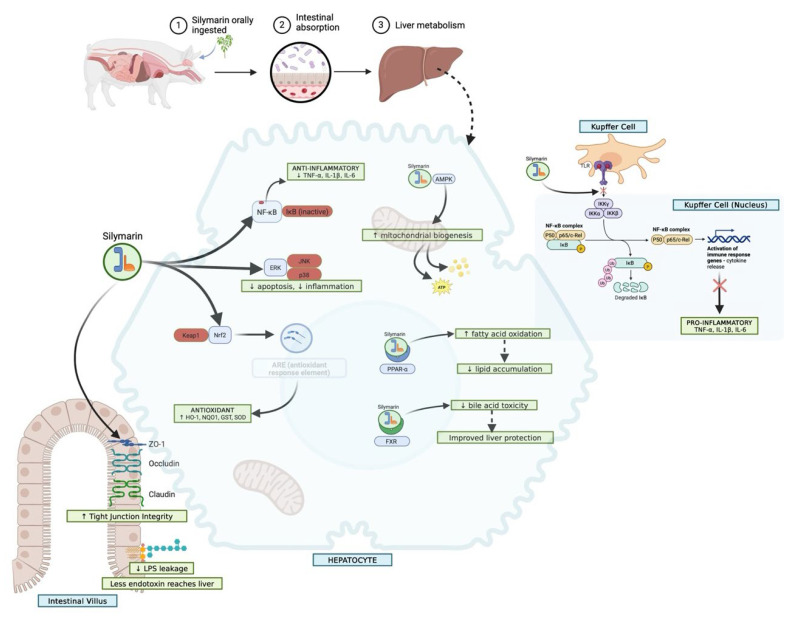
Hepatoprotective and gastroprotective mechanisms of silymarin. This schematic representation delineates the pharmacokinetic trajectory and molecular pathways that underpin the hepatoprotective properties of silymarin. Subsequent to oral administration (1), silymarin is subjected to intestinal absorption (2) and hepatic biotransformation (3). Within the intestinal environment, silymarin fortifies tight junction integrity through the upregulation (↑) of occludin, claudin, and ZO-1, consequently mitigating lipopolysaccharide (LPS) leakage (↓) and restricting endotoxin translocation to the hepatic tissue. In hepatocytes, silymarin activates the Nrf2 pathway by disrupting the Keap1–Nrf2 interaction, which induces the transcription of antioxidant response elements (ARE) and the subsequent upregulation of antioxidant enzymes (HO-1, NQO1, GST, SOD), collectively attenuating oxidative stress. Concurrently, silymarin inhibits (×) pro-inflammatory signaling pathways mediated by NF-κB, ERK, JNK, and p38 MAPK, thereby diminishing apoptosis and inflammatory responses. The activation of AMPK promotes mitochondrial biogenesis, thereby enhancing energy metabolism, while the stimulation of PPAR-α and FXR pathways facilitates fatty acid oxidation, reduces lipid accumulation, and alleviates bile acid toxicity, culminating in augmented hepatocellular protection. In Kupffer cells, silymarin modulates TLR/NF-κB signaling pathways, obstructing the nuclear translocation of the NF-κB p65/p50 complex and inhibiting the expression of pro-inflammatory cytokines (TNF-α, IL-1β, IL-6). Collectively, these synergistic actions across the intestinal, hepatic, and immune systems establish silymarin as a multi-faceted hepatoprotective agent, exerting its effects through antioxidant, anti-inflammatory, mitochondrial, and barrier-stabilizing mechanisms.

**Figure 5 nutrients-17-03278-f005:**
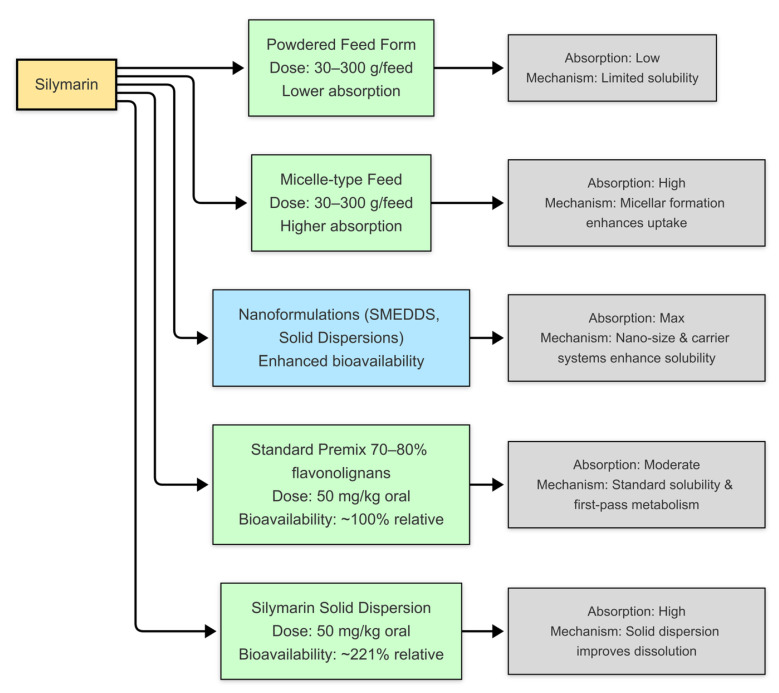
Comparative Bioavailability of Different Silymarin Formulations. Silymarin shows notoriously poor oral absorption in its raw powdered form because of low solubility, but formulation technologies have been developed to overcome this limitation. Micelle-based feeds improve uptake by packaging flavonolignans into micellar structures that are more easily absorbed. Nanotechnology approaches such as self-micro emulsifying systems (SMEDDSs) and solid dispersions take this further, greatly enhancing solubility and maximizing systemic exposure. Standard premix formulations, typically containing 70–80% flavonolignans, achieve moderate absorption but remain affected by first-pass metabolism. In contrast, optimized solid dispersions improve dissolution enough to raise relative bioavailability by more than two-fold (~221%). These differences underline how strongly the clinical effectiveness of silymarin depends not only on dose but also on the formulation strategy used.

**Table 1 nutrients-17-03278-t001:** Comparative hepatic physiological and pharmacokinetic features of pigs and rodents in relation to silymarin studies.

Parameter	Pig Liver	Rodent Liver	Relevance to Silymarin Studies
Lobular architecture	Hexagonal lobules with comparable central vein organization and sinusoidal spacing	Smaller lobules with denser sinusoidal packing	Reflects a similar intrahepatic distribution of hepatoprotective agents [27]
Sinusoidal hemodynamic	Continuous low-pressure perfusion; fenestrated endothelium mimicking human sinusoid	Higher flow velocity; less fenestration	Allows accurate modeling of microcirculatory oxidative stress [28]
Bile-acid composition	Mixed glycine- and taurine-conjugated bile acids	Predominantly taurine-conjugated	Closely reflects human bile-acid-dependent drug metabolism [29]
Transporter repertoire (OATP, NTCP, BSEP, MRP2)	Expression patterns parallel human hepatocytes	Distinct isoforms and substrate affinities	Enables realistic modeling of enterohepatic recirculation and detoxification [30]
Cytochrome P450 profile	CYP3A, CYP2E1, and CYP1A2 similar to human expression ratios	Overrepresentation of the CYP2C family	Predictive for the oxidative biotransformation of polyphenols [31]
Hepatic immune microenvironment	Kupffer cell activity, TLR4 response patterns mirror humans	Hyper-reactive cytokine profile	Crucial for assessing redox–inflammatory modulation by silymarin [32]
Pharmacokinetic translatability	Comparable bioavailability and half-life kinetics	Rapid metabolism, low plasma retention	Improves dose-to-effect extrapolation for clinical nutrition models [33]

This table summarizes key hepatic and pharmacokinetic parameters comparing porcine and rodent models, emphasizing their translational relevance to silymarin research. The porcine liver closely mirrors human hepatic physiology across multiple dimensions—lobular organization, sinusoidal hemodynamic, bile-acid composition, transporter expression (OATP: organic anion-transporting polypeptides; NTCP: sodium taurocholate transporting polypeptide; BSEP: bile salt export pump; MRP2: multidrug resistance-associated protein 2), cytochrome P450 enzyme distribution (CYP3A, CYP2E1, CYP1A2), and immune microenvironment—making it a robust model for evaluating hepatoprotective compounds. These similarities enhance the predictive value of porcine studies for pharmacokinetic translation, dose–exposure extrapolation, and mechanistic assessment of redox–inflammatory modulation by silymarin.

**Table 2 nutrients-17-03278-t002:** Principal flavonolignans of milk thistle (*Silybum marianum*) and their molecular/functional characteristics.

Compound		IUPAC Name/Alternative Name(s)	Molecular Formula/Weight	Structural Features	Relative Abundance in Silymarin Extract	Pharmacological Relevance
Silibinin (Silybin)	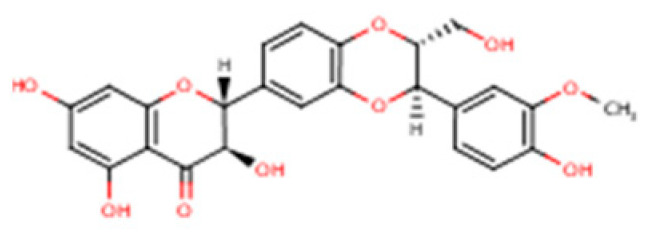	(2R,3R)-3,5,7-trihydroxy-2-(3-hydroxy-4-methoxyphenyl)-2,3-dihydro-4H-chromen-4-one lignan conjugate; occurs as silybin A and silybin B	C_25_H_22_O_10_/482.44 g·mol^−1^	Major flavonolignan diastereomeric pair, taxifolin nucleus + coniferyl alcohol	50–70% of total silymarin	Dominant hepatoprotective agent; antioxidant (Nrf2/ARE activation), antifibrotic (TGF-β1 blockade), cytoprotective against xenobiotics; basis for silibinin-phytosomes formulations
Isosilybin (diastereomeric pair)	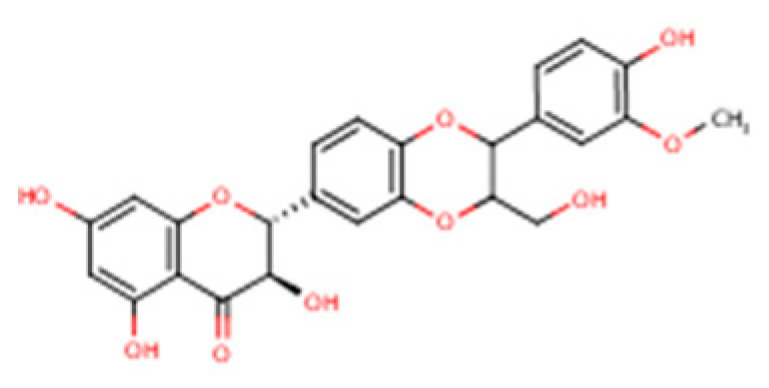	Diastereomeric forms of silibinin (Isosilybin A and Isosilybin B)	C_25_H_22_O_10_/482.44 g·mol^−1^	Stereoisomeric variation at C-10 and C-11	5–10%	Exhibits antioxidant and anti-neoplastic activity (ERK1/2, Akt modulation); less abundant but pharmacologically distinct
Isosilybin A	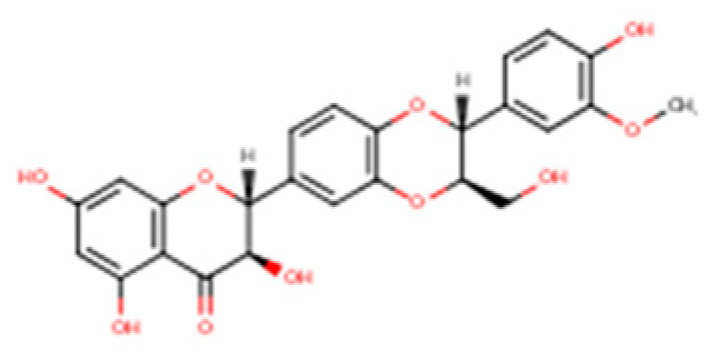	(2R,3S)-diastereomer of Isosilybin	C_25_H_22_O_10_/482.44 g·mol^−1^	Stereospecific structural variant	Minor fraction	Reported cytoprotective effects in hepatocytes; selective anti-cancer potential (induces G1 arrest, modulates p53 signalling)
Silychristin	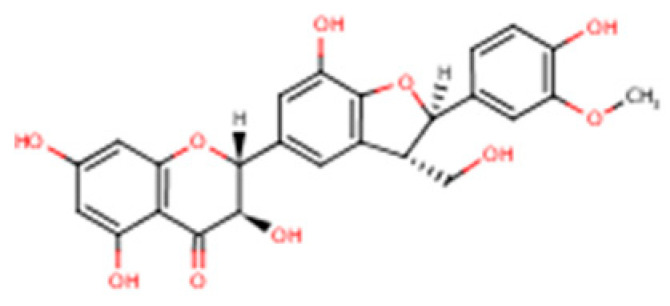	(2R,3R)-flavonolignan with guaiacyl unit; also termed silychristin A	C_25_H_22_O_10_/482.44 g·mol^−1^	Flavonolignan with a unique ether linkage; second most abundant after silibinin	20–25%	Mitochondrial protective, potent antioxidant, suppresses oxidative phosphorylation imbalance; anti-apoptotic, anti-inflammatory
Isosilychristin	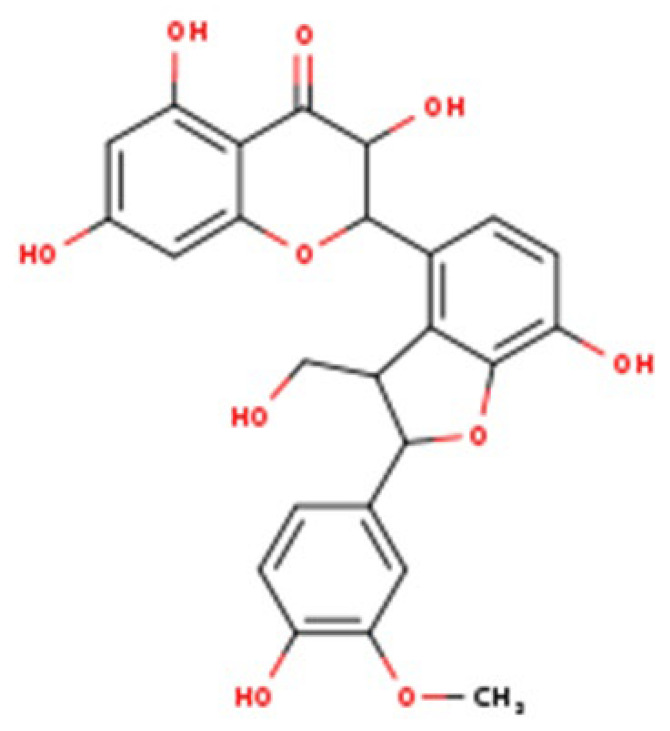	Stereoisomer of silychristin	C_25_H_22_O_10_/482.44 g·mol^−1^	Chiral variant of silychristin	<5%	Limited pharmacological data; evidence for antioxidant and mild hepatoprotective actions
Silandrin	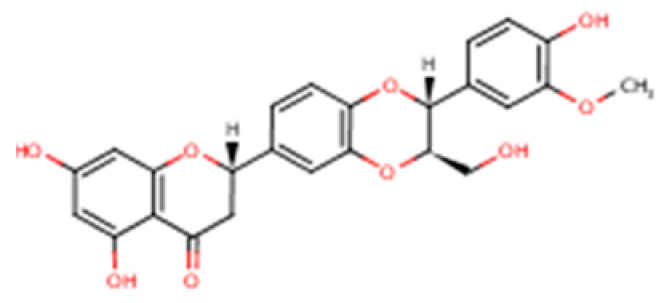	Rare minor flavonolignan derivative	C_25_H_22_O_10_/482.44 g·mol^−1^	Structural analogue with an altered linkage pattern	Trace (<1%)	Weak hepatoprotective potential; limited experimental validation compared to major flavonolignans

**Table 3 nutrients-17-03278-t003:** Therapeutic effects and mechanistic pathways of silymarin and its flavonolignan constituents.

Property/Effect	Description	Molecular Mechanisms	Representative Outcomes	Citation(s)
Antioxidant defense	Augments cellular redox buffering and suppresses ROS/RNS accumulation under toxic and metabolic stress.	Direct scavenging of OH, O_2_ and HOCl; Nrf2–ARE pathway activation with transcriptional induction of HO-1, NQO1, γ-GCS, and GSH-synthetase; stabilization of Keap1–Nrf2 interface; suppression of COX-1/2 and 5-LOX catalytic activity.	Upregulation ↑ of GSH, SOD, CAT (20–40%); MDA and lipid peroxidation; preserved mitochondrial redox potential under APAP, CCl_4_, and HFD challenge.	[13,56,57,58]
Anti-inflammatory activity	Blunts innate and adaptive pro-inflammatory signalling.	Inhibition of NF-κB p65 nuclear translocation; suppression of MAPKs (ERK1/2, JNK, p38); downregulation of COX-2, iNOS, 5-LOX; attenuation of NLRP3 inflammasome activation in Kupffer cells.	IL-1β, IL-6, TNF-α (30–50%); IL-10; CRP; reduced hepatic neutrophil/macrophage infiltration and parenchymal necro-inflammation.	[56,59,60]
Immunomodulation	Fine-tunes T-cell-mediated immunity in a dose-dependent manner.	Low-dose: inhibition of T-cell proliferation; suppression of NF-κB and ERK1/2/p38 MAPK in CD4^+^ subsets; modulation of mTOR–S6K signalling; restoration of Th1/Th2 equilibrium. High-dose: paradoxical stimulation of pro-inflammatory cascades.	IL-2, IFN-γ, TNF-α by 30–60%; restoration of Th1/Th2 balance; potential dose-dependent duality in adaptive immunity.	[13,61,62]
Cardiovascular protection	Reduces ischemic injury and preserves myocardial contractility.	PI3K/Akt activation and GSK-3β phosphorylation; NF-κB inhibition; stabilization of sarcolemmal and mitochondrial membranes; attenuation of Ca^2+^ overload; modulation of CXCR4/SDF-1 axis in cardiomyocyte repair.	infarct size (50–85%); CK-MB, troponin I, LDH; restored MAP and ejection fraction; arrhythmia incidence.	[63,64,65,66]
Hepatoprotection	Guards hepatocytes against xenobiotic, metabolic, and viral insults.	Membrane stabilization; inhibition of TNF-α/TNFR1 and TGF-β1/Smad2/3 pathways; stimulation of ribosomal RNA polymerase I and ribosome biogenesis; competitive blockade of hepatotoxin uptake transporters (OATPs).	ALT/AST; fibrosis indices; cytochrome c oxidase; caspase-3 activity; DNA fragmentation; improved histopathological scores.	[13,45,60,67]
Metabolic regulation	Improves insulin sensitivity and reprograms hepatic lipid metabolism.	Restoration of IRS-1/PI3K/Akt signalling; FXR activation and SHP induction; suppression of NF-κB in adipose macrophages; modulation of ACC, FAS, CPT-1, and SREBP-1c.	HOMA-IR 15–25%; fasting glucose; visceral fat and BW; leptin and NPY; PYY.	[68,69,70]
Anti-fibrotic action	Suppresses hepatic stellate cell activation and extracellular matrix deposition.	Inhibition of TGF-β1/Smad2/3 transcription; downregulation of α-SMA; NF-κB inhibition; reduced MCP-1 and Ly6C^high monocyte recruitment; blockade of HSC–myofibroblast transition.	collagen deposition 40–60%; α-SMA expression; TGF-β1; improved Masson’s trichrome fibrosis scores.	[59,71]
Anti-neoplastic	Exerts cytostatic and pro-apoptotic effects across diverse tumour models.	Fas/FasL-mediated extrinsic apoptosis; upregulation of Bax/Bcl-2 ratio; PARP cleavage; p53 stabilization; inhibition of ERK/MAPK signalling; arrest at G_0_/G_1_ and G_2_/M checkpoints.	tumour cell viability 40–70%; apoptotic index (13–67%); tumour volume (~26% in vivo); caspase-3 activation.	[72,73,74]
Lipid-lowering	Corrects dyslipidaemia and promotes bile acid metabolism.	Inhibition of HMG-CoA reductase; upregulation of LDLR and CYP7A1; enhanced fecal bile acid excretion; modulation of CYP4A and PPARα pathways.	total cholesterol (20%); TG (23%); ↓ LDL-C (18%); VLDL-C (26%); HDL-C.	[75]
Neuroprotection	Protects neurons from ischemic and oxidative injury.	PI3K/Akt activation; induction of BDNF; upregulation of Nrf2/HO-1 axis; inhibition of p53, Bax, caspase-3; preservation of mitochondrial membrane potential.	infarct volume 30–50%; Nrf2/HO-1; neuronal apoptosis and GFAP; improved cognitive/behavioral tests.	[76,77,78,79]

## Data Availability

Not applicable.

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
