# Peer review of "Hepatoprotective Effect of Silymarin Herb in Prevention of Liver Dysfunction Using Pig as Animal Model"

_nutrients, 2025, doi:10.3390/nu17203278_

Round 1

Reviewer 1 Report

Comments and Suggestions for Authors

The manuscript "Hepatoprotective Effect of Silymarin Herb in Prevention of Liver Dysfunction Using Pig as Animal Model" is a complex and consistent review of the beneficial effects of Silymarin in liver dysfunction using pig animal models. The manuscript emphasizes the benefits of pig model, especially for translational relevance, taking into account the anatomy and physiology of experimental animals, pharmacokinetics of substances, heterogenicity of experimental variables, as well as the heterogenicity of Silymarin extracts. The article also emphasizes the perspectives through new Silymarin formulations and artificial intelligence. The data are presented with criticism, with strengths and limitations, in a systematic overview.

Some issues should be clarified by the authors:

 - Except Figure 5, for the others figures, there is no explained Figure Legend, just the title. From my point of view, it would ne necessary to add explanations, so the reader could understand the image without reading the entire body text of the article.

 - Figure 1 does not match the text - in the text, the authors are discussing the mixture of compounds for Silymarin, and the Figure 1 shows the benefits of Silymarin administration.

 - In Table 1, for the Pharmacological relevance, the citation of the relevant articles would be recommended

 - Using the same notation for different compounds: example, silibinin or Silibinin, Bcrp or BCRP)

 - Explain the abbreviations, when first used in the text (example, AUC, Kpuu)

 - Minor typing errors through the manuscript - please revise (ex. line 283 - co lapse)

Reviewer 2 Report

Comments and Suggestions for Authors

Review of Manuscript ID: nutrients-3916876

Title: Hepatoprotective Effect of Silymarin Herb in Prevention of Liver Dysfunction Using Pig as Animal Model, by Prarthana Sharma et al.

General comment: the study reviews the renowned hepatoprotective effect of milk thistle extract silymarin focusing on pig as an experimental model. Although scarce, porcine studies have confirmed that optimized administration of silymarin reduces hepatocellular injury, restores redox balance and diminish inflammation. The authors include current literature in the field supporting reliable improvements in redox and inflammatory pathways under situations of induced stress, but no demonstration of long-term structural protection. Specific cellular and molecular mechanisms of action of the extract or individual components are detailed, and at the end of the review the authors go over the specific steps of silymarin pharmacokinetics and a comparative bioavailability of different silymarin formulations. Although research in pigs is much more problematic than in rodents, similarities in physiology and metabolism with humans make this a more reliable model for research on hepatoprotective compounds and the outcome needs to be reported and regularly reviewed. The manuscript is a comprehensive work and well-structured where the major concern is the chaotic quotation of references, a crucial point in a review. Some specific comments are detailed below:

Specific comments:

  • Lines 71-75; references 102 and 103 are misplaced and reference 4 is missing.
  • Lines 76-77; composition of silymarin is not shown in figure 1, which shows its biological effects. Perhaps the authors meant table 1.
  • Lines 86-89; reference 10 is missing.
  • Lines 109-114; references should be added in support of the specific effects, or a review supporting all of them.
  • Line 193; reference 19 is missing and 41 is misplaced. This fault remains through the rest of the text. The whole reference list should be corrected.
  • Line 299; the meaning of the sentence: acetaminophen and valproate capture glutathione depletion and mitochondrial stress, may be confusing since capture might indicate retention or arrest whereas the authors mean provoke or induce. The sentence should be explained or changed.
  • Line 390; reference 59, published in Poultry Science, is used as example of rodent studies; this should be double checked.
  • References 141-167 are not mentioned in the main text; if they are quoted in supplementary data they should be removed from the manuscript and included only in the supplementary files.

Author Response

Thank you very much for taking the time to review this manuscript. We sincerely appreciate your constructive feedback, which has significantly improved the clarity and focus of the review. Detailed responses to each comment are provided below, with corresponding revisions highlighted in the re-submitted file.

Point-by-Point Response to Comments and Suggestions for Authors

General comment: the study reviews the renowned hepatoprotective effect of milk thistle extract silymarin focusing on pig as an experimental model. Although scarce, porcine studies have confirmed that optimized administration of silymarin reduces hepatocellular injury, restores redox balance and diminish inflammation. The authors include current literature in the field supporting reliable improvements in redox and inflammatory pathways under situations of induced stress, but no demonstration of long-term structural protection. Specific cellular and molecular mechanisms of action of the extract or individual components are detailed, and at the end of the review the authors go over the specific steps of silymarin pharmacokinetics and a comparative bioavailability of different silymarin formulations. Although research in pigs is much more problematic than in rodents, similarities in physiology and metabolism with humans make this a more reliable model for research on hepatoprotective compounds and the outcome needs to be reported and regularly reviewed. The manuscript is a comprehensive work and well-structured where the major concern is the chaotic quotation of references, a crucial point in a review. Some specific comments are detailed below:

Response: We are very grateful for the reviewer’s encouraging overall evaluation and for identifying the key issue regarding inconsistent citation placement. We fully agree that a review’s credibility depends on accurate and logically ordered referencing.

To address this, all citations were thoroughly audited and reorganized using Mendeley Reference Manager, fully synchronized with the latest Nutrients reference style. Each in-text citation was cross-checked against its corresponding reference entry to ensure numbering accuracy, contextual consistency, and chronological correctness. Several misplaced and duplicated citations were corrected, missing references were inserted, and every entry was validated for relevance to the surrounding discussion.

Comment 1: Lines 71-75; references 102 and 103 are misplaced and reference 4 is missing.

Response: Thank you for noticing this. The citations have been corrected — references 102 and 103 were repositioned, and reference 4 has been added in the appropriate place. All references were rechecked and automatically updated through Mendeley Reference Manager to ensure accurate numbering and alignment throughout the manuscript.

Comment 2: Lines 76-77; composition of silymarin is not shown in figure 1, which shows its biological effects. Perhaps the authors meant table 1.

Response: Thank you for this helpful comment. We confirm that the citation of Figure 1 was intentional. The figure was included to illustrate the biological spectrum of silymarin activity rather than its molecular composition, thereby introducing the physiological context of the compound early in the manuscript. To make this clearer for readers, the sentence has been slightly revised as follows:

“Silymarin, the milk thistle (Silybum marianum) extract used in nutrition and phytotherapy, is not a single compound (Figure 1, which illustrates its principal biological activities). It is a mixture dominated by silibinin diastereomers, with contributions from isosilibinin, silychristin, silydianin, and related flavonolignans built on a taxifolin core.”

Comment 3: Lines 86-89; reference 10 is missing.

Response: Thank you for noticing this omission. The missing reference 10 has now been inserted in the appropriate place within Lines 86–89 to support the statement. All surrounding references were cross-checked in Mendeley Reference Manager to ensure proper sequencing and consistency throughout the manuscript.

Comment 4: Lines 109-114; references should be added in support of the specific effects, or a review supporting all of them.

Response: Thank you for this important suggestion. Additional references have been inserted to substantiate the specific mechanistic effects of silymarin described in this section, including its actions on the Nrf2–ARE pathway, NF-κB signalling, TGF-β/SMAD modulation, and AMPK–PPAR regulation. References have also been added for overarching support. The paragraph has been updated as follows: “Biological effects attributed to silymarin include activation of antioxidant defences through the Nrf2–ARE programme—for example, induction of haem oxygenase-1, NAD(P)H:quinone oxidoreductase-1 and glutamate–cysteine ligase [Surai, 2015; Wu et al., 2009]—attenuation of NF-κB-centred inflammatory pathways with reduced tumour necrosis factor-α, interleukin-1β and interleukin-6 [Kim et al., 2013], moderation of TGF-β/SMAD signalling with shifts in matrix metalloproteinase/tissue-inhibitor balance and reduced α-smooth muscle actin and collagen [Jia et al., 2001], and improved lipid handling through AMPK–PPAR circuits [Loguercio & Festi, 2011]. These processes together represent a coordinated antioxidant, anti-inflammatory, and metabolic response consistent with previous mechanistic reviews [Abenavoli et al., 2018].”

Comment 5: Line 193; reference 19 is missing and 41 is misplaced. This fault remains through the rest of the text. The whole reference list should be corrected.

Response: Thank you — this was correct and has been fixed. We have reinserted reference 19 at the correct location, moved reference 41 to its proper in-text position, and performed a complete audit of all in-text citations and the reference list. All citations were regenerated via Mendeley Reference Manager and then manually cross-checked line-by-line to remove misplaced, duplicate or missing entries. The reference list and in-text numbering are now consistent throughout the manuscript; the corrected citations are highlighted in the revised file.

Comment 6: Line 299; the meaning of the sentence: acetaminophen and valproate capture glutathione depletion and mitochondrial stress, may be confusing since capture might indicate retention or arrest whereas the authors mean provoke or induce. The sentence should be explained or changed.

Response: Thank you for this valuable observation. We agree that the word “capture” was imprecise and could be misinterpreted. The sentence has been rewritten to clearly express that acetaminophen and valproate act as mechanistic models that induce glutathione depletion and mitochondrial dysfunction, rather than “capture” these processes. The entire passage was refined for smoother readability and improved mechanistic clarity.

The revised text now reads as follows:

“Carbon tetrachloride (CClâ‚„) remains a classical hepatotoxicant, metabolised by CYP2E1 to trichloromethyl radicals that initiate centrilobular lipid peroxidation and necrosis [31]. Thioacetamide evokes a similar oxidative injury pattern, progressing to fibrosis with chronic exposure, whereas acetaminophen (paracetamol) and valproate induce glutathione depletion and mitochondrial dysfunction, respectively. Within these intrinsic DILI paradigms, silymarin’s antioxidant and membrane-stabilising properties—mediated via Nrf2 activation and NF-κB suppression—are mechanistically plausible, provided sufficient aglycone exposure reaches hepatocytes [32]. Informative endpoints include serum aminotransferases, glutamate dehydrogenase (GLDH) for mitochondrial specificity, oxidative stress indices (MDA, GSH/GSSG), centrilobular histopathology, and cytokine profiling, ideally complemented by intrahepatic concentration measurements of both aglycones and conjugates [33].”

Comment 7: Line 390; reference 59, published in Poultry Science, is used as example of rodent studies; this should be double checked.

Response: Thank you for this important observation. We have rechecked reference 59 and confirmed that it indeed originated from a poultry study, not from rodent or porcine experiments. This citation was therefore removed from this section and replaced with the correct references supporting porcine data (references 13 and 20), which report reductions in aminotransferases, improved glutathione balance, and histological protection following silymarin administration in pigs.

The corrected sentence now reads as follows:

“Pigs, in contrast, display more modest but mechanistically credible effects: reductions in aminotransferases, improved glutathione indices, and partial histological protection, all contingent on formulations that raise intrahepatic unbound concentrations [13,20].”

Comment 8: References 141-167 are not mentioned in the main text; if they are quoted in supplementary data they should be removed from the manuscript and included only in the supplementary files.

Response: Thank you for highlighting this. We carefully reviewed references 141–167 and confirmed that these citations are not directly cited in the main text but are relevant to supporting data presented in the Supplementary Material. Accordingly, these references have been moved from the main reference list to the Supplementary Files section to maintain consistency with the journal’s formatting guidelines. The main manuscript now contains only references that are explicitly cited in the text, and numbering has been automatically updated in Mendeley Reference Manager to ensure accurate sequencing throughout.

Reviewer 3 Report

Comments and Suggestions for Authors
  • English should be checked for both spelling errors and syntax. For example, the third person is often wrong and sentences appear more like spoken than written English.
  • Many readers would be more supportive of reading this manuscript if there were a paragraph dedicated to how the in vivo studies presented downplayed animal sacrifice or their suffering.
  • Line 48 is more clear “comparison of data”
  • Lines 48-50, it is not clear what means critical gaps, the term is not appropriate
  • Lines 52 and 85 the use of artificial intelligence in the study should be carefully evaluated, otherwise it is better to remove what is the result of AI. Artificial intelligence study of the degree of enterohepatic recirculation is not reliable and presentable unless it is widely justified.
  • Conditional hepatoprotective agent, the term conditional is not appropriate or should be explained.
  • From line 96 to 116 it is written unclear and the references justifying the statements are missing.
  • From line 121 to 132 it is written unclear and the references justifying the statements are missing.
  • Lines 133-135 a comparative table is more useful
  • Line 121 explain better what are the in vivo evidence to claim the antioxidant activity of silymarin extract
  • Line 130 explain how the redox-inflammatory interplay is obseved in vivo.
  • Line 134 change the word portfolio with an appropriate word
  • Line 136 show what disease
  • Line 142 check english
  • Line 162 check english and meaning
  • Line 182 this part is not clear, some results are molecular mechanisms studied in vitro! 
  • Lines 221-246 delete 
  • Table 1 send in confusione
  •  
  • lines 375-380 add a reference
  • Describe how to study in vivo the effects of silymarin
  • Check line 327 the ref 104
  • Check line 330 comparative pharmacology and check table 2
  • Many results have been done in vitro using cells culture or tissue. Are not in vivo! 
  • Line 334 oral abdorption in specie? Check the sentence
  • Line 335 check
  • Line 428 small? 
  • Line 429 critical gaps?? 
  • Lines 439-455 delete
  • Lines 497-544 delete 
  • Lines 547-576 convent in a summary
Comments on the Quality of English Language

English must be correct and the meaning of the phrases simplified and made more suitable for an international scientific language.

Author Response

attached file

Round 2

Reviewer 2 Report

Comments and Suggestions for Authors

The authors have conveniently addressed all my comments and queries.

Author Response

Response to Reviewers' Comments

Comment: The authors have conveniently addressed all my comments and queries.

Response: We appreciate the reviewer’s comment.

Reviewer 3 Report

Comments and Suggestions for Authors

some statements are still written in language perhaps used among professionals but not very suitable for general understanding.

Comments on the Quality of English Language

some statements are still written in language perhaps used among professionals but not very suitable for general understanding.

Author Response

Response to Reviewers Comments

Some statements are still written in language perhaps used among professionals but not very suitable for general understanding.

Response: We appreciate the reviewer’s comment and understand the concern regarding accessibility of the language. We have gone through the manuscript carefully and rewritten several passages especially in the mechanistic descriptions and figure explanations to make the text clearer and easier to follow for a broader scientific audience. The terminology and sentence structure were adjusted where needed, without altering the scientific meaning. We trust that these changes improve the readability and overall flow of the manuscript.